# Why Are You Turning a Blind Eye to Fair Trade Coffee?—Focused on the Comparison between Korea and Africa

**Mina Jo** [1] , **Haggai Kennedy Ochieng** [2] **and Jisong Kim** [3,*]

1    Division of Hotel & Tourism, College of Economics & Business Administration, The University of Suwon, Hwaseong 18323, Republic of Korea
2    Department of Liberal Arts, Duksung Women's University, Seoul 01369, Republic of Korea
3    Department of Economics, College of Economics & Business Administration, The University of Suwon, Hwaseong 18323, Republic of Korea
*    Correspondence: jisongkim@suwon.ac.kr; Tel.: +82-31-229-8238

**Abstract:** This study examines the reasons why people turn a blind eye to fair trade coffee (FTC) and the way to revitalize the FTC market. In-depth interviews with eight Koreans and eight Africans living in Korea were conducted and analyzed qualitatively. Our thematic analysis employed NVivo Word Cloud analysis and Text Search to analyze the relationship among words. The study reveals that Koreans do not purchase FTC due to low awareness, low priority among coffee selection attributes, low accessibility and low product competitiveness. As producers, Africans do not produce FTC because of a lack of information, low returns, cultural factors, lack of interest from political leaders and corruption. African consumers do not purchase FTC due to low accessibility, low priority among coffee selection attributes, doubts about FTC and its high price. To revitalize the FTC market on the demand side, it is important to improve accessibility, promote and market the brand and pay attention to cultural constraints.

**Keywords:** fair trade coffee (FTC); ethical consumption; Korea; Africa; qualitative research

## 1. Introduction

Coffee is one of the most consumed beverages around the world and the global demand is anticipated to increase, particularly in non-traditional coffee drinking countries such as Africa, Asia, and Oceania [1]. According to estimates, by 2030, the global demand for the beans will increase to 200 million bags [2]. For coffee-producing developing countries, coffee production is a major source of livelihood. In Africa, it is one of the most important export crops, generating a significant part of national income and a vital source of foreign exchange earnings. Over 38% of the population of Burundi, 23% Tanzania, 22% Uganda, 17% Côte d'Ivoire and 14% Ethiopia depend on coffee production [3]. Despite being the origin of coffee, Africans rarely consume coffee [2]. In 2021, Africa accounted for only 7.4% of global coffee market [4]. This pattern is gradually changing with the emergence of domestic coffee shops. International coffee brands such as the Starbucks Corporation, Nestle NESCAFE, and Chameleon Cold Brew are also slowly setting a foothold in Africa [5]. For Africans embracing coffee, it represents modernization. Coffee brands are positioning it as a new lifestyle and everyone who wishes to appear elegant and fashionable identifies with coffee drinking.

South Korea, on the contrary, is an established coffee market with a long history of coffee. The coffee market in Korea has grown rapidly. Koreans consume 353 cups of coffee per person per year, which is 2.67 times the global average. It is estimated that Korea has 70,000 cafés [6]. With coffee having a strong footprint in Korea and with coffee being known to be a unique part of Korean identity, coffee consumers are giving a new value to coffee brands by using it as a means to express their individuality or self [7]. As the number of consumers who want value for a cup of coffee and improved quality of life increases,

the overall perception of the emotional benefits from the service and cafe atmosphere has become the core of the marketing strategy [8].

Moreover, coffee is one of the ethically branded products in the global market. There is an increasing number of consumers who value ethical consumption, focusing on places of true value rather than material satisfaction. Ethical consumption can be defined as the act of building products or services of value, in social or environmental terms, rather than as competitive alternatives in terms of price and quality [9]. One of the areas receiving the most attention among ethical consumers is fair trade [10]. Fair trade pursues greater equity in international trade and is defined as a trade partnership based on respect and transparency [11]. According to the Fairtrade Labeling Organizations International (FLO), global fair trade sales grew 16% year-on-year, to approximately $7.3 billion in 2015 [12]. Coffee forms the core of the fair trade movement and is the most widely consumed fair trade product in European markets and is quickly expanding in other markets [13]. The period between 2004 and 2014 experienced a significant growth of other ethical coffee accreditation schemes, such as the Rainforest Alliance and UTZ. Many coffee companies, such as Starbucks, have already integrated the Corporate Social Responsibility (CSR) into their business models [14]. The acceleration in the development of the coffee market means consumers should no longer struggle to find a coffee brand that has some form of ethical accreditation.

Despite steady growth and interest, awareness of fair trade coffee is still low. In South Africa, for instance, despite significant market expansion in fair trade products, as well as being one of the mature coffee markets, research found that only 7% of their sample of 1507 consumers were aware of Fairtrade as an organization, and 6% were familiar with the logo [15]. However, the number of fair trade organizations in Africa has grown from 444 (2016) to 596 organizations (2018) [16]. Low awareness of fair trade leads to a low purchase rate of fair trade coffee [17]. There are a number of studies regarding fair trade coffee in Korea and African countries [18–23]. Some studies from Korea and Africa have shown that awareness of the fair trade concept increases the consumers' purchasing intention for fair trade-compliant products, as well as increase customer satisfaction [15,24]. However, research on fair trade coffee in Korea and Africa is limited compared to western countries, where relatively diverse studies on fair trade coffee consumption have been conducted. As such, it is still unclear which factors inhibit the growth of fair coffee markets in Korea and Africa. This study aims to fill this gap by investigating cross-cultural reasons why coffee consumers ignore fair trade coffee (FTC), which is considered to be ethical consumption. In addition, the study examines ways to revitalize FTC markets to promote ethical consumption in both Korea and Africa. The study conducted qualitative analysis based on an in-depth interview with eight Koreans and eight Africans. With regard to the consumption of fair trade coffee, the study found that in both Korea and Africa, coffee consumers do not buy fair trade coffee due to low accessibility and low priority among coffee selection attributes. As a coffee-producing region, Africa does not produce fair trade coffee because of a lack of information regarding fair trade coffee, low returns, and cultural factors that inhibit them from growing non-traditional crops such as coffee.

## 2. Literature Review

### 2.1. The Coffee Market

Korea has grown into the world's sixth-largest coffee consumer market after Europe, the US, and Japan. Currently, it is a huge market, worth 5.7 billion dollars [25]. Coffee imports (coffee beans, green beans, etc.) reached 168,000 tons [26], and the domestic coffee market was anticipated to exceed 5.7 billion dollars in 2018 [16]. Recently, as a consequence of the growth of coffee shops, consumers' perceptions have changed significantly and the number of people who enjoy coffee at home is increasing [27]. Franchise coffee shops are forming a new trend in the coffee market, with low-priced coffee, which is going head to head with expensive premium coffee [18]. Moreover, the emergence of private coffee shops

featuring high-end interiors and premium coffee is noticeable with the entry of global coffee companies into the domestic market [28].

The coffee industry continues to show a rapidly changing trend, with "specialty coffee" becoming popular and "decaffeinate coffee" in the spotlight. As Koreans' tastes become more sophisticated, coffee consumers are increasingly considering the quality of coffee beans and origin, and a special coffee craze using specialty high-end beans is evolving [29]. According to a 2017 survey conducted by Jadeng on a coffee bean company, 51% of consumers choose "taste and aroma" when purchasing coffee, while 28% of consumers consider "price" [30]. Kim [31] classified the selection attributes in franchise coffee shops in Korea. The study shows that important performance factors include friendliness, coffee prices, cleanliness of the restroom, the quick response to an order by staff, the freshness of the coffee, the indoor ambience, the diversity of the menu, business hours, appealing indoor interior design and staff knowledge regarding products and coffee taste. Kim [32] found that, if the amount spent on coffee per month among college students is less than 30,000 won, low-cost coffee is preferred, and coffee quality is not considered. Nevertheless, as the size of the allowance increases, coffee quality becomes a consideration. On the other hand, Kim [33] considered 'diversity of coffee' to be very important when the motivation to visit coffee shops is an economical or habitual factor, and 'modulate price of coffee' to be important when it comes to convenience or habitual motivation.

The domestic coffee market in Africa is still in its infancy, with the exception of Ethiopia, which has a traditional coffee culture, with 50% of the production sold in the domestic market [34]. South Africa is another mature coffee-drinking market with many brands of coffee [35]. The low coffee consumption culture in Africa is gradually changing and presents plenty of opportunity for growth in the domestic coffee market. This is reflected in the emergence of local coffee shop chains, such as Nigeria's Neo Café, Ethiopia's Kaldi's and Kenya's Art Caffe and Java House. Dorman's is the dominant local player, with a presence along the entire coffee value chain, from regional bean sourcing to roasting and retail [2]. International coffee franchises are not widespread in Africa, but there is a creeping presence of major international brands including Starbucks, Nestle NESCAFE, Coca-Cola, and Chameleon Cold Brew. Starbucks opened its first store in South Africa in 2016 and has expanded its portfolio in South Africa and other African countries [5].

Between 2016 and 2020, coffee consumption in Africa grew by 2.1%, compared to Europe: 1.9%, Asia and Oceania: 1.3%, Mexico and Central America: 0.6%, North America: 3.7% and South America: 1.1% [4]. The growth in the domestic coffee market is driven by an emerging middle class that is stimulating demand for consumer goods, including coffee [36]. The majority of coffee consumers are urban and city dwellers who have more disposable income [37]. Coffee consumption is also influenced by exposure and particularly Western influence [24]. Branding coffee as the new lifestyle drink has also been a vital influence in people converting to coffee [38].

The characteristics of coffee consumers are heterogeneous across Africa. This heterogeneity is evident in age groups, preferred places for drinking coffee, purpose, and even gender. For example, in Nigeria, coffee is seen as a social drink that is consumed in social places over meetings, whereas in Uganda, 71.3% is home-based consumption. The majority of coffee consumers in Nigeria are young people, while in Uganda the majority of consumers fall between 30 to 49 years. Most people in Africa consume coffee primarily as a means to keep warm, but coffee chain stores are also positioning it as a better way for consuming caffeine. Statistics further show that coffee consumers in some African countries (Ethiopia 55%, Nigeria 53%, Kenya 50%, Tanzania 50% and Ghana 40%) are willing to pay more for higher quality coffee or tea [39].

### 2.2. Ethical Consumption: Fair Trade Coffee

Ethical consumption encompasses practices such as the Fair Trade movement, the Clean Clothes Campaign, boycott, and animal welfare movements [40]. Consumers typically express their ethical concerns by purchasing products that have positive attributes

or by boycotting products that have negative attributes [24]. A growing body of literature documents that ethical consumption is rising in Korea and Africa [41,42]. South Africa is the fastest growing fair trade market in Africa, followed by Kenya [43]. A wide range of FT products—coffee, tea, wine, chocolate, herbs—are already available in local stores in South Africa. The first South-to-South FT model in Africa was launched in 2009 with Label South Africa (FLSA) as a marketing organization. In 2013, the Fairtrade Marketing Organization Eastern Africa (FMOEA) was launched in Kenya as a marketing organization for FT-labeled products in Eastern Africa [44]. Since then, the FT market has recorded tremendous growth. Similarly, with the growth of the domestic coffee shop market in Korea, the perception that fair trade coffee equates with good coffee has expanded, and a number of fair trade coffee shops have emerged.

The rise in ethical consumption in Korea and Africa is driven by activism by NGOs to mobilize ethical consumers as well as CSR. The initiatives combine globalizing business and political networks of responsibility with local institutions to stimulate ethical consumption [19]. Whitehead et al. [45] explain that retailer stores also play a vital role in shaping ethical consumption. In South Africa, Pick n Pay, the largest South African FT vendor, recorded a 182% growth in coffee sales in 2014 from the figures of 2013, while Woolworths recorded a 153% increase in sales of FTC [46]. Similarly, a wide array of locally produced FTC, Cadbury Dairy Milk made with Fairtrade cocoa and sugar produced in Africa is already selling in Kenyan retail stores.

A similar trend is observable in Korea, where many retail stores sell ethically branded products, with the pursuit of ethical consumption by the Consumer Cooperatives of Korea (CCK) growing rapidly [41]. As competition in the coffee market intensifies, the pursuit of existing profits and CSR are becoming important as a means of discriminatory marketing to improve brand image and secure competitiveness. At the same time, individual ethical consumption has also become a hot topic. CSR and individual consumption became a hot topic in the coffee market when the International Coffee Agreement collapsed, leading to a sharp fall in coffee prices in 1989, hitting a bottom low in 2002 [47]. Consequently, large multinational corporations came under intense pressure from NGOs, leading to the emergence of CSR. Starbucks, P&G, Kraft, SLDE and Nestle, among others, purchase fair trade coffee or produce high-quality coffee at a reasonably high price through direct transactions with coffee farms in developing countries [14].

Fair trade coffee is a new product in African countries and in other countries in which coffee less consumed; therefore, it may not have formed consumers' purchase preferences [48,49]. Yang et al. state that consumers' awareness and purchasing intentions for fair trade coffee are closely related to coffee consumption [50]. In countries with high coffee consumption, such as Korea, the fair trade coffee market is also well established. Rizqiyanto revealed that most consumers who purchase fair trade products in these countries are people who have the awareness of ethical consumptions [17].

## 3. Research Methodology
### 3.1. Method

The study adopted a qualitative research design. An in-depth interview method was used to conduct the research. In-depth interview is a qualitative research technique to conduct intensive individual interviews about a particular idea, program or situation with a small number of respondents. The in-depth interview method is useful when you seek to obtain detailed information about a person's in-depth thoughts or actions on a particular topic [51]. The aim of this study was to explore the cross-cultural reasons why coffee consumers ignore fair trade coffee (FTC), which is considered to be a form of ethical consumption. In addition, the study examines ways to revitalize the FTC market to promote ethical consumption in both Korea and Africa. Therefore, an in-depth interview method was chosen to carry out the research.

### 3.2. Participants

Amongst its Korean participants, the study examines consumption behaviors from the perspective of coffee consuming country. Amongst its African participants, the study is intended to investigate their behaviors from the perspective of coffee-producing countries. Cognizant of the limitations of interviewing respondents from a single country, participants from different African countries were selected for the interviews.

The snowball sampling method was used to recruit interview participants. As the research was conducted during COVID-19, it was hard to meet or reach the Korean and African interviewees. As a result of conducting a pilot test before starting the main interview, it was revealed that people's general awareness and experience of fair trade coffee was too low. Therefore, we conducted in-depth interviews with Korean and African students and graduates who took fair trade-related classes or had some prior knowledge of fair trade coffee. The Korean interviewees were those who had taken classes related to fair trade coffee, worked as a barista in the coffee industry, or had experience working at FTC companies.

The African interviewees currently live in Korea for various reasons, such as studying or working. However, those targeted for the interview must have been born and lived in Africa for more than 20 years. Those who declined an interview because they did not know much about fair trade coffee were excluded from the interview. The researchers obtained informed consent from the interview participants before conducting the interviews. The researchers explained the purpose of the study, targeted participants, inter-view procedures, study period, potential risks and discomforts, benefits, withdrawal from the study and/or withdrawal of authorization, confidentiality, compensation, voluntary participation and authorization. If the interview participants agreed to the above condition, the participant's signature was received. The interviews stopped when there were no more new topics to discuss concerning FTC and they became saturated. The final sample was 16 people: eight Koreans and eight Africans participated in the interviews, respectively.

### 3.3. Data Collection

Individual qualitative semi-structured interviews were conducted from July 1st to August 21st, 2021. An interviewer from Africa interviewed the African respondents because he shared a similar cultural background and could easily build a rapport with the African interviewees. All of the African interviews were conducted in English because the interviewer and the respondents are fluent in English. The interviews with Koreans were conducted in Korean by a Korean-American who is fluent in Korean and was also in a position to build a rapport with the Korean interviewees. Only one interviewee was present in each interview session. The interviewers have experience and adequate training in conducting qualitative interviews. The mean time for the interviews was 58 min and ranged between 39 and 74 min. The interviews were conducted via Zoom video conference. (https://zoom.us/, accessed on 27 July 2021). Each interview was recorded with the consent of the interviewee and transcribed verbatim by using the Daglo internet website (https://daglo.ai/, accessed on 26 July 2021). Daglo is a service that converts voice recordings into text using artificial intelligence (AI) technology. A research assistant then individually checked the errors in the transcription program and corrected them. All the authors checked the transcribed files again.

### 3.4. Analysis

Prior to the analysis, the transcribed data were shown to the participants to check whether there were any errors with the data. For data analysis, initial coding was performed and the transcription contents were read repeatedly for verification. Subcategories and categories were derived through within-group comparison and cross-group comparison using a constant composite method. The results analyzed by the first author were reviewed repeatedly by the second and third authors, and opinions on the analysis results were coordinated to ensure reliability and trustworthiness.

The researchers used the NVivo R program and Text Search Query software for qualitative analysis. NVivo has been used in other studies to analyze interview data [52]. NVivo is a software developed by QSR International for qualitative data analysis, such as content analysis and narrative analysis. The software provides a workplace for researchers to store, manage, query, and analyze unstructured data, including text, images, audio, video, and other data types. NVivo also allows users to complete multiple qualitative analysis functions on the platforms, including sorting and filtering raw data, discovering and building relationships among data, assigning and defining themes and categories for data, visualizing data analysis results, and creating reports [51].

Themes and concepts were derived from the data by inductive process. Whilst the NVivo R program is a useful analysis tool when analyzing qualitative data, it does not automatically extract themes and concepts from the data. However, the program is useful for the visualization of the qualitative data analysis results.

This study used thematic analysis, one of the qualitative research methodologies. Comparative analysis was conducted between the answers from the Korean participants, as a coffee-consumers and citizens of a developed country, and the answers from the African participants as a coffee-producing and developing region. The NVivo R program and Text Search Query software were used in the qualitative analysis to confirm the relationship between each word by combining simple words, such as stemmed words, synonyms, specializations, and generalizations. Text Search Query searches for words or phrases specified in the data, and a subset of the data. The researchers chose to use Text Search Query because it is necessary to locate all related data with the "fair trade coffee" key phrases. Text Search Query provides a single window with all references that satisfy the criteria. One visual output is the Word Tree that presents recurrent phrases in a branching structure, which is known as key-word-in-context (KWIC). Lastly, a project map was used to refine the coding hierarchy. It can show the structure of coding the system in diagrammatic form in a project map [53].

## 4. Findings

The analysis of this study reveals the reasons for the lack of consuming and producing FTC for Koreans and African, and the ways to revitalize the FTC market as a consumer or producer country. The researchers used NVivo R to explore and present the links between project items. NVivo is a qualitative data analysis software package and is designed for qualitative researchers working with text-based information. In this paper, the researchers used a text search query, which provided visualizations by retrieving all the words or phrases that satisfy the criteria by searching for 'fair trade coffee' as a keyword. The text search query results show the reasons why Koreans do not buy fair trade coffee (Figure 1), the reasons why Africans do not produce fair trade coffee (Figure 2), and the reasons why Africans do not buy fair trade coffee (Figure 3).

A project map providing a graphic representation of the different items is used. To create the project map, the researchers drew a map showing the relationship between each interviewee and the corresponding responses. The researchers then added other associated items to help illustrate the results. The code that is more connected to the interviewee is more important because it is the content that more interviewees responded to. The project map shows the interview results on how to revitalize the FTC market for Koreans as a consumer (Figure 4), for Africans as a producer (Figure 5), and Africans as a consumer (Figure 6).

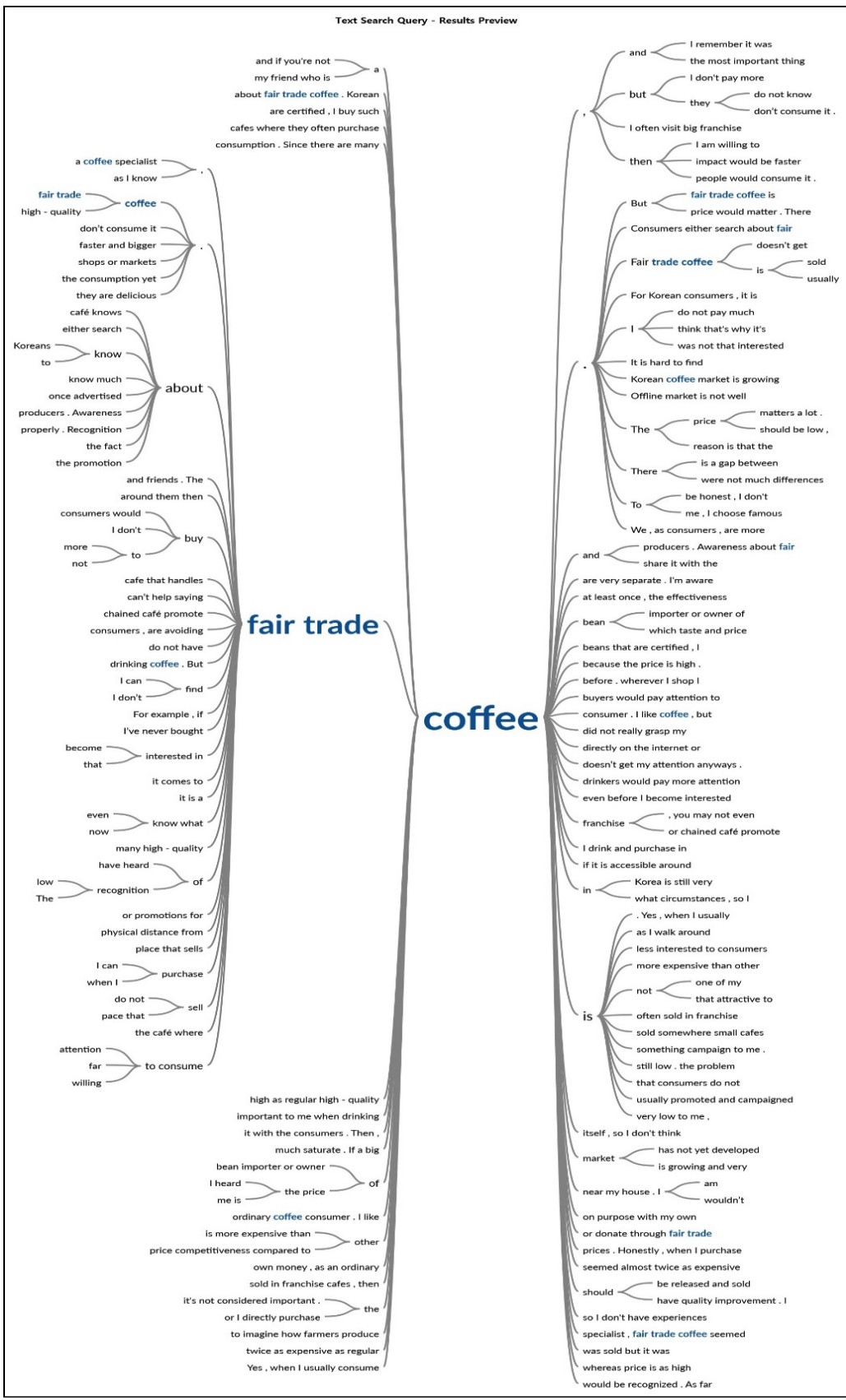

**Figure 1.** The reasons Korean consumers do not buy fair trade coffee.

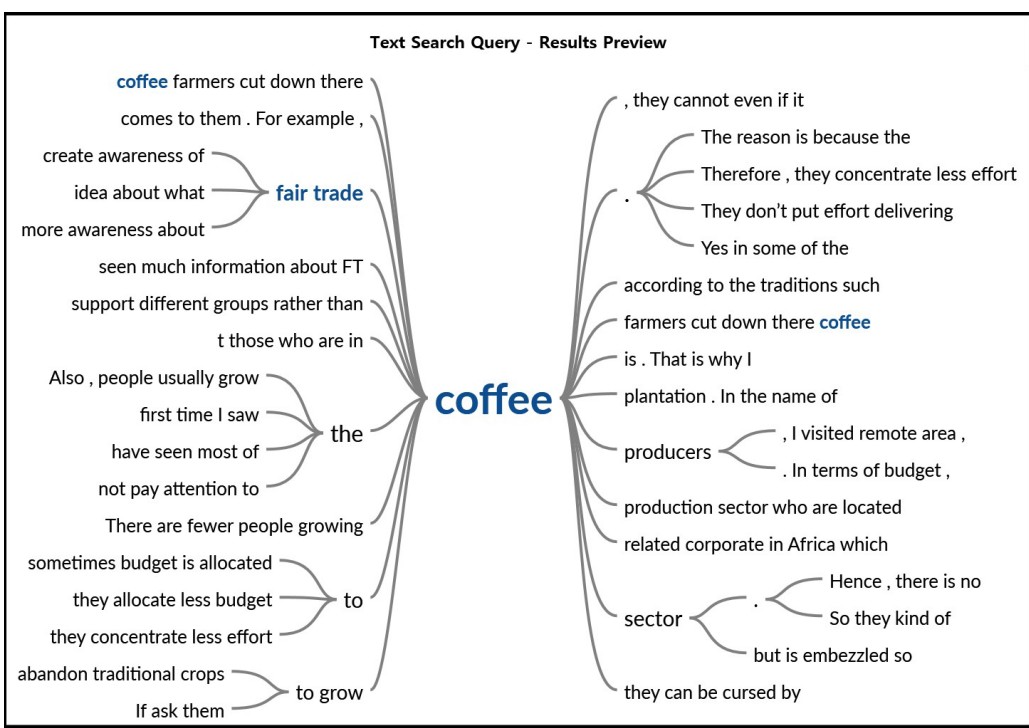

**Figure 2.** The reasons African producers do not produce fair trade coffee.

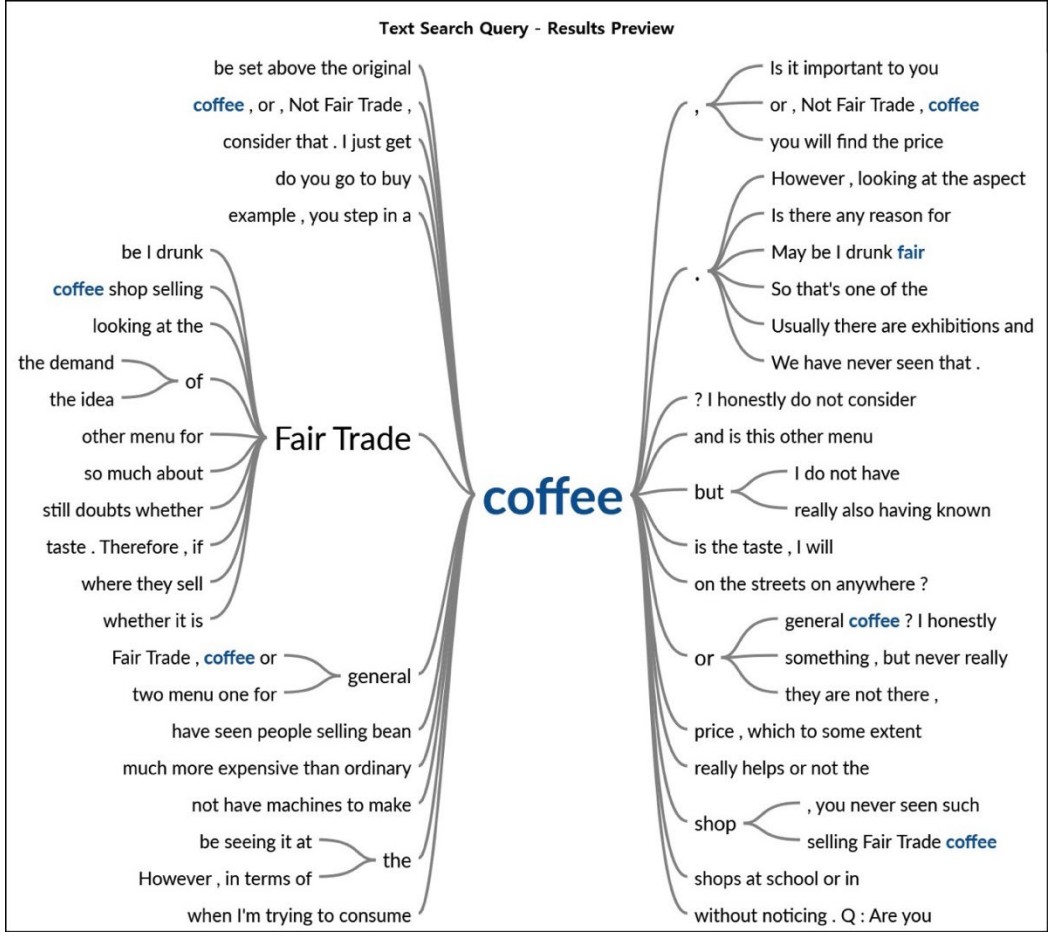

**Figure 3.** The reasons African consumers do not buy fair trade coffee.

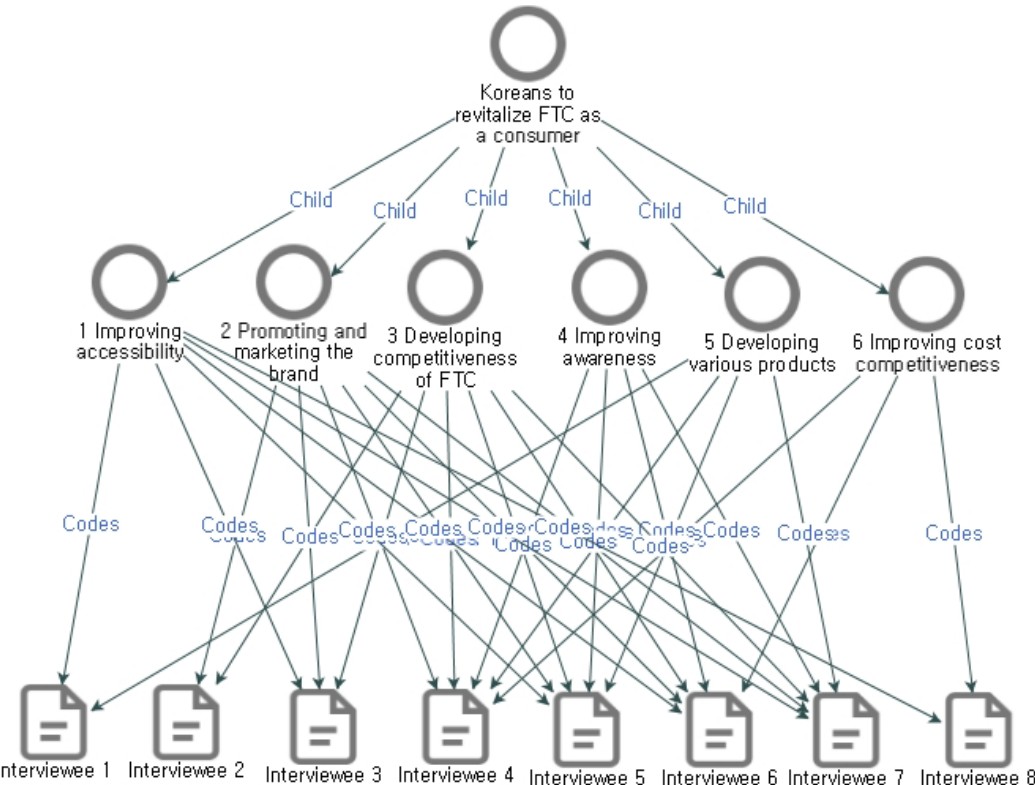

**Figure 4.** Project map for Koreans to revitalize FTC market as a consumer.

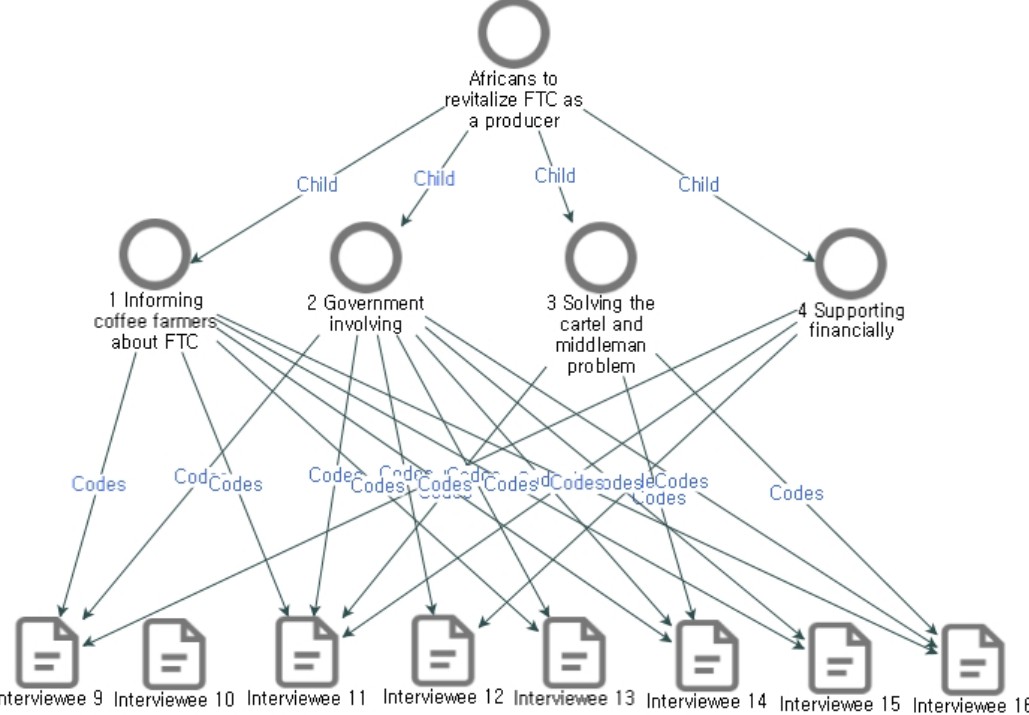

**Figure 5.** Project map for Africans to revitalize FTC market as a producer.

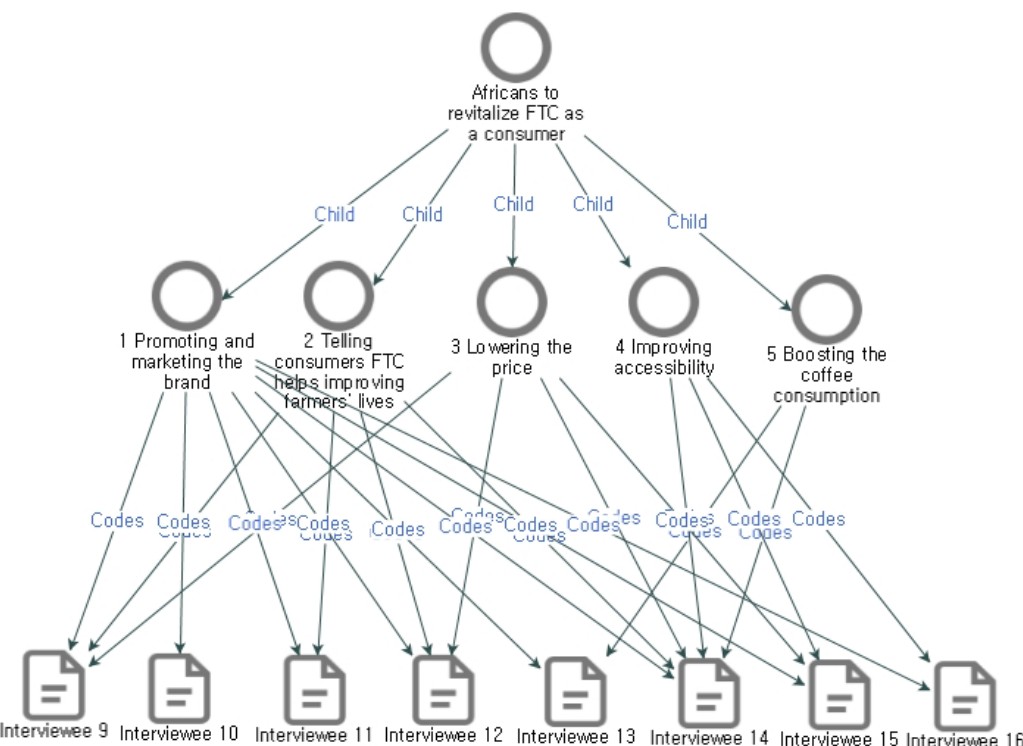

**Figure 6.** Project map for Africans to revitalize FTC market as a consumer.

*4.1. Reasons for Not Consuming and Producing FTC*

4.1.1. Case of Koreans

Consumer Side

According to the analysis, the reasons why Korean consumers do not consume Fair trade coffee include: 'low level of awareness'; 'low priority among coffee selection attributes'; 'low accessibility'; 'low product competitiveness compared to price'; 'doubts about FTC'; and 'consumers' uninterestedness'. It is noteworthy from the analysis that despite interviewees majoring in FTC-related majors, such as international development cooperation or hospitality, their awareness of fair trade coffee is low. In most cases, the research participants indicated that they have never heard of FTC, and even if they major or work in related fields, they only have a shallow idea of the concept, and do not know about it. Koreans assign low priority to FTC among coffee selection attributes because Korean coffee consumers value taste and price most when choosing coffee. In addition, Koreans place higher priority on the interior or atmosphere of the coffee drinking location. As a result, even though the purpose of the FTC is meaningful, it possesses low priority among the various attributes of choosing coffee. Regarding low accessibility, the respondents indicated that it is difficult to find chain stores that sell FTC, but when they are available, the location is often too far and difficult to access when one is not deliberately looking for FTC. Regarding the low product competitiveness compared to price, Korean respondents contend that FTC is expensive, yet ordinary in terms of quality of taste. This implies that FTC's product competitiveness is low. The Korean coffee market has witnessed tremendous growth, with high-quality specialty coffees increasing. Compared to specialty coffee, which has an excellent taste and aroma, FTC is only expensive. This means consumers do not make repeated purchases of FTC because of its poor quality. Consumers can purchase a few cups of FTC out of sympathy or alignment with its good purpose. Nonetheless, for consumers to develop loyalty to the brand and make repeat purchases, the FTC must be competitive. Regarding doubts about FTC, the participants pointed out that even if consumers intended to purchase FTC for the ethical purposes, such as improving the lives of farmers, they were reluctant to do so because of unstable inventory and low performance

reviews from existing customers. With regard to consumers' 'uninterestedness', the Korean respondents were found to be quite indifferent to the FTC. Although some coffee chains such as Starbucks SPC, one of the biggest coffee franchises in Korea, advertise FTC, participants explained that their awareness is decreasing because they are not interested in FTC itself.

"For Korean consumers, it is still a beginning stage to know about fair trade coffee. When I was learning about fair trade in school, I asked around and only a few people knew about it." (Interviewee 3).

"To be honest, I don't buy fair trade coffee on purpose with my own money, as an ordinary coffee consumer. I like coffee, but I do not pay more to buy fair trade coffee. The reason is that price and taste are important to me when drinking coffee. However, fair trade coffee is not that attractive to me. It does not come as a charm when it comes to taste. Am I going to pay that much for it? I cannot decide about that." (Interviewee 8).

"I learned about fair trade coffee in school, and lately I have started to pay attention to it. But … it was hard to find it somewhere near my neighborhood. It was difficult to find a place where I could purchase fair trade coffee. Offline market is not well developed yet in Seoul." (Interviewee 5).

"It is definitely taste and price competitiveness. No matter how expensive it is, I will buy it if it tastes good, and people will buy it if there is price competitiveness. I can't help but talk about quality improvement. If it's tasty and cheaper, people will choose it." (Interviewee 2).

"Honestly, when I purchase fair trade coffee or donate through fair trade products, I think there is a lack of confidence that it is going to producers properly." (Interviewee 4).

"Fair trade coffee is less interesting to consumers because they do not find it important. If barista or person who works at café knows about fair trade coffee and share it with the customers, then coffee buyers would pay attention to it." (Interviewee 2).

4.1.2. Case of Africans
Producer Side

The study shows that the reasons why African producers do not produce FTC are as follows: 'lack of information about FTC'; 'low returns leading to change to alternative crops'; 'cultural factors'; 'lack of interest from political leaders and government corruption'; and 'cost requirements'. Regarding the lack of information regarding FTC, coffee farmers often had no idea about FTC. As most of the interviewees had heard about fair trade coffee for the first time in Korea, they did not know exactly the difference in the profit distribution structure between ordinary coffee and FTC. In addition, in rural Africa, there are many places without electricity; thus, unless farmers deliver coffee beans directly to the market or are linked to cooperatives, communication with the outside is cut off, making it difficult to gain information regarding FTC. Low returns and turning to alternative crops are due to distrust that even if fair the FTC generates premiums, much of it only benefits the middlemen or retailers who are not directly involved in coffee production, rather than the actual producers. In terms of cultural factors, African farmers have not fully embraced changes such as FTC because they are accustomed to traditional crops they have cultivated from generation to generation. Some farmers believe that if traditional crops are abandoned, they can be cursed by their ancestors. Regarding the lack of interest from political leaders and government corruption, the participants explained that there are countries with various social and economic problems, such as civil war, dictatorship, and inequality. As political leaders invest government resources in other cash crops that can maximize their own profits, there is a lack of support for FTC to improve the lives of coffee farmers. In respect to cost requirements, the respondents explained that overhead costs, such as FTC certification requirements, are prohibitively costly to poor famers in Africa, particularly for small-scale farmers.

"It is very low among Africans, and generally, in rural areas in developing countries. The reason is that I have not seen much information about the FTC. Governments do not

put effort into delivering the information, and elites are keeping it to themselves so that the benefits can only go to them. For example, coffee related cooperatives in Africa are mostly run by the elite, who only care about their own benefits." (Interviewee 16).

"There are fewer people growing coffee. This is because the returns are low . . . Because they are peasants, they end up growing just common crops for their sustainability. Some farmers inter-crop by planting avocado and other common crops like corn potato and beans." (Interviewee 10.

"Yes, in some of the areas, they grow some crops for prestige. If you ask them to grow coffee, they cannot, even if it is profitable. They believe that if they abandon traditional crops to grow coffee they can be cursed by ancestors." (Interviewee 9).

"I can talk of political leaders in my country. Because they are involved in production of different cash crops like maize and so on, they do not pay attention to the coffee sector . . . So they make policies based on what benefits them first. In this way, there is conflict of interest. The leaders support different groups rather than coffee producers. Hence, there is no budget to give experts to create awareness of fair trade coffee . . . Maybe sometimes, there is corruption because sometimes budget the is allocated to coffee sector, but it is embezzled so it is not going to do the original intention." (Interviewee 9).

"Implementing the strategy requires money. It requires resources of time and finances, which are sometimes not available. This hinders effective implementation." (Interviewee 13).

Consumer Side

The study further shows that the reasons why Africans, as consumers, do not consume fair trade coffee include: 'low accessibility'; 'low priority among coffee selection attributes'; 'double about FTC'; and 'high price'. Regarding low accessibility, most of the African interviewees (including those from coffee-farming families) had never heard of FTC before coming to Korea. Even once in Korea, they only saw it at exhibitions, and in many cases, they had never encountered fair trade coffee in actual coffee shops. Regarding low priority among coffee selection attributes, price and taste are the important factors when choosing coffee, while FTC is not considered to be an important selection attribute for the interviewees. According to the African interviewees, people who did not drink coffee in Africa began to drink coffee while in Korea. However, they mainly prefer soft-tasting coffee loaded with sugar and milk. Most of them do not see coffee as a social drink as the practice is not rooted in their culture. As such, they visit coffee shops and use coffee as a social drink only when they are in Korea. However, when in Africa, they often drink powdered coffee or canned coffee, considering the price and taste. In respect to doubts about FTC, the interviewees indicated they do not believe that FTC was helpful to farmers. Moreover, there is widespread distrust that even if profits were generated from FTC, political leaders, middlemen and global companies monopolized the profits, and it would not benefit farmers. In relation to high price, it is a common principle that demand decreases when prices increase, according to the law of supply and demand. Thus, although FTC is a noble idea, it is questionable whether it can reverse the trend, particularly when considering low-income consumers in Africa. In Africa, coffee consumption is generally low because many Africans do not see value in spending money on coffee, which is prohibitively expensive, when they can drink tea at a substantially lower cost.

"I think the reason is that it is not popular. Not so many people know about it. For example, you step into a coffee shop; you never see FTC on the menu. If it was popular, I am sure I would be seeing it at the coffee shops at school or in my city, but I have never seen that." (Interviewee 14).

"Basically, a consumer looks for maximum utility, and the main focus for me is the taste." (Interviewee 10).

"There are some people who know about it but do not believe in it. I think there are still doubts as to whether fair trade coffee really is helpful to farmers. Does it really protect

the environment? Is it for everyone whether you are rich or poor? There are these kind of doubts in Senegal." (Interviewee 11).

"I don't understand, why should you put the price to be higher than the other coffee? Is the consumer to bear the cost, or is it that the businesspeople are making too much profit, which they should share with the farmer. Fair trade is a noble idea, but in terms of increasing the price to the consumer, I do not understand how the concept works." (Interviewee 10).

"I know that it is usually much more expensive than ordinary coffee. So that's one of the reasons." (Interviewee 13).

### 4.1.3. Revitalizing of FTC Market
#### Korean FTC Market

According to the Korean interviewees, the following factors are critical in revitalizing the Korean FTC market: 'improving accessibility'; 'promoting and marketing the brand'; 'developing competitiveness of FTC'; 'improving awareness'; 'developing various products'; and 'improving cost competitiveness'. In terms of providing accessibility, it is necessary to form a coffee brand chain in Korea to collaborate with coffee shops run by large companies, such as Starbucks, to increase opportunities to taste FTC, or to include FTC in coffee menus to increase exposure. Moreover, the respondents pointed out that FTC is less accessible because coffee shops that currently handle FTC sell exclusively it in the form of beans, whilst Koreans usually consume coffee as a drink at cafes rather than in the form of beans. This means that accessibility can be improved when it is sold as a beverage. In terms of promoting and marketing the brand, the interviewees explained that young people should actively promote FTC by providing fun and friendly media content using various SNS. Regarding the developing competitiveness of FTC, participants highlighted that consumers should not be forced to embrace ethical consumption. Furthermore, it is important to ensure that the quality of FTC matches the price, which is often high. Making FTC a sophisticated and desirable brand would generate demand and simultaneously improve ethical consumption. Regarding improving awareness of FTC, the respondents explained that priority should be given to improving the image of companies dealing with FTC. Only then can the recognition of FTC increase. From the perspective of developing various products, if countries importing FTC were to diversify, then consumers would be able to experience products of various tastes and quality. In addition, it is necessary to develop various products by not only developing coffee in the form of beans, but also in other forms, such as powdered or canned coffee. Concerning cost improvement, it is necessary to lower the FTC price. The interviewees explained that the current price of FTC is not competitive. The price of coffee beans is an important issue not only for individuals who consume coffee drinks, but also for owners who run coffee shops. For both consumers and sellers, FTC prices are unreasonably expensive, which implies the pricing structure should be improved.

"When I lived in the US... I regularly went to a mart that only sells organic products.... There were a variety of organic and fair trade products, so I often bought one or two there. In Korea, if I could find fair trade coffee at coffee shops I often go to, I would buy them." (Interviewee 8).

"In order for many people to know about FTC, you have to let them recognize it easily and be familiar with it. You can hold many events using Instagram and YouTube or introduce them to blogs on portal sites such as Naver and Daum. If you upload content that includes the concept of fair trade coffee, it will be easy and fun for many people to enjoy." (Interviewee 5).

"People who want to buy fair trade coffee are those who already know about fair trade. In the case of British brand Lush, consumers do not purchase the product for ethical consumption. As the product itself is so attractive, it can also instill a perception that I seem to be making ethical consumption. Is there a fair trade brand that consumers want to consume in Korea?" (Interviewee 2).

"The image of companies engaged in fair trade coffee-related businesses should be improved. If those companies' images improve, wouldn't it be a natural way for people to know that they are doing a fair trade coffee business when their initials are included?" (Interview 4).

"I don't think it has to be coffee beans, and if possible, wouldn't there be a way to develop and sell powdered coffee?" (Interviewee 1).

"It is important to target people in their 20s to 40s who consume the most coffee and making more drinks that can be enjoyed in various ways." (Interviewee 5).

### 4.1.4. African FTC Market
#### Producer Side

The interview with the respondents revealed the following ways to revitalize African FTC producers: 'informing coffee farmers about FTC'; 'government involvement'; 'solving the cartel and middlemen problem'; and 'supporting financially'. There can be several ways to inform farmers about FTC: first, Farmers can be educated through local cooperative or village meetings. Second, by providing education directly to coffee farmers. Finally, they can be educated using mass media such as TV, radio, and newspapers. As most African farmers are poor and excluded from education and information networks where they can learn about FTC, direct education to farmers would be a better alternative. It is also desirable to actively utilize local cooperatives or village meetings. As cooperatives are activated by region, the latter method can be more efficient than approaching individual farmers. Among mass media, respondents explained that channels such as TV, radio and newspapers are the most frequently used media by farmers in rural areas, which makes them the ideal tools for communication with farmers. In terms of government involvement, the respondents elucidated that the role of government is important in increasing FTC production. The government should allocate an adequate budget to the coffee sector, provide subsidies, educate farmers and help them acquire certification. The government should equally involve themselves in eliminating cartel networks and coffee middlemen who have captured the coffee sector. However, the respondents expressed concerns about the opacity of inter-governmental transactions and their connections with middlemen. This is consistent with Ingenbleek and Reinders [54], who found that the lead companies, namely, retailers and coffee roasters, significantly influence decisions related to sustainable coffee market. This breeds doubts about FTC. Cole and Brown [48] similarly raised concerns that producers have little voice within the system, even though FTC is designed to provide them social, economic and environmental benefits. With regard to financial support, the respondents explained that the cost to acquire FTC licenses for poor farmers in Africa is too high. Thus, in order to increase FTC production, the cost of acquiring FTC licenses should be lowered or subsidized by the responsible agencies to increase the uptake of FTC farming. This is in line with Haight [49], who found that strict certification requirements lead to uneven economic advantages among coffee growers and a lower quality for consumers.

"Definitely, reach out to more and more farmers. Once these farmers see one farmer benefiting, they encourage their fellow farmers to join the cooperatives." (Interviewee 13).

"Using radio would be a good example. Almost everyone in Africa especially in rural areas connects to the radio station. Passing some small messages through the SNS would work in passing information to the farmers." (Interviewee 16).

"...the famers who are producing coffee should get a better price. The government should regulate companies buying coffee products from farmers. They should set a price relative to the price at which these buyers will resell in the world market. Or there should be an auction for coffee like they do with other crops in Tanzania, where the highest bidder determines the price at which the crop is sold." (Interviewee 9).

"Farmers are not motivated because there is involvement of a lot of middlemen. That really prevents farmers from getting what they deserve. If it were possible to remove those middlemen, farmers would get involved in coffee production. It would be great in Senegal and in Africa in general." (Interviewee 11).

"To be certified as a fair trade coffee producer, there is a registration cost that you have to incur to get a license, which is not less than $5000. To African countries, this is a lot of money . . . The government should lower registration cost for those dealing in fair trade coffee." (Interviewee 9).

Consumer Side

The respondents explained the following ways to revitalize the African FTC consumer market as follows: 'promoting and marketing the brand'; 'telling consumers FTC helps improve farmers' lives'; 'lowering the price'; 'improving accessibility'; and 'boosting the coffee consumption'. From the perspective of promoting and brand marketing, the interviewees were of the opinion that it is necessary to inform more consumers by actively promoting FTC. 'Telling consumers that FTC helps improve farmers' lives' was found to be important as respondents asserted that African consumers are familiar with and sympathize with African rural farmers. Thus, they are willing to consume FTC if they are convinced that purchasing FTC helps improve farmers' lives. Moreover, lowering the price of FTC is also an important factor in revitalizing the African consumer market. According to the participants, no matter how noble the purpose of supporting the producer is, there is a maximum amount that consumers are willing to pay. As a result, it is necessary to adjust the price appropriately to stimulate demand. In terms of improving accessibility, FTC promoters should implement marketing strategies such as providing free samples to accord buyers the opportunity to taste the products and increase brand awareness. Schollenberg [55] estimated the hedonic price for FTC in Sweden and contended that a premium of 38%should be paid for FT-labeled coffee. However, this estimate could vary between countries. Additionally, it is vital to increase accessibility in terms of opening more coffee shops, as well as making FTC available and easily recognizable in coffee shop menus. As for boosting coffee consumption, the participants mentioned that for consumption to be activated, coffee consumption culture must be addressed. Once coffee consumption is entrenched in consumer culture and people embrace coffee, such as in Korea, consumers will be interested in other types of coffee, such as FTC. It is a point worthy of note that Africans do not have a coffee consumption culture, and the practice is still in its infancy.

"Maybe if there is a way to motivate coffee shops to display fair trade coffee or even to make some posts that can inform people about fair trade coffee. Hence, I think more needs to be done to advertise it. The first thing is to make sure to attract more coffee shops, advertise it, make people know about that." (Interviewee 14).

"I want to know that my money is contributing, even if in a small way, to making their lives better. The environment too . . . You have to show them how just buying small cup of fair trade coffee contributes to changing the farmer's life and the communities, and protecting the environment." (Interviewee 11).

"The challenge is to make sure the price is balanced because even if you want to support the producer, there is a minimum amount or maximum amount the consumer is willing to pay." (Interviewee 14).

"When you want to make your product known to the public, the first thing you do is to share it with people. You can try to make some free fair trade coffee and share some samples with people to taste and to be aware that something like that exist the market." (Interviewee 15).

"If people are serious about promoting FT coffee, then it is important to make an agreement between the agent and the coffee shops to show the label about the FT coffee. When customer goes to the coffee shop, they should be able to find the FT coffee in the menu." (Interviewee 16).

## 5. Conclusion and Implications

### 5.1. Conclusions

Fair trade coffee is coffee with a good purpose of promoting ethical consumption and enabling poor farmers in developing countries to increase returns from production.

However, even Korea, with high coffee purchases, and Africa, a coffee producer, are less aware of the FTC and rarely purchase or produce FTC. Therefore, this research examined the reasons why coffee consumers and producers do not consume and produce FTC despite its noble intentions, and how to revitalize the FTC market.

The results of this study show that Koreans do not consume fair trade coffee due to 'low level of awareness'; 'low priority among coffee selection attributes'; 'low accessibility'; 'low product competitiveness compared to price'; 'doubts about FTC'; and 'consumers' uninterestedness'. On the other hand, as producers, Africans do not produce fair trade coffee due to 'lack of information about FTC'; 'low returns leading change to alternative crops'; 'cultural factors'; 'lack of interest from political leaders and government corruption'; and 'cost requirement'. As a consumer market, Africans do not consume fair trade coffee due to 'low accessibility'; 'low priority among coffee selection attributes'; and 'doubts about FTC and high price'. To revitalize the Korean FTC market, the respondents suggested 'improving accessibility'; 'promoting and marketing the brand'; 'developing competitiveness of FTC'; 'improving awareness'; 'developing various products'; and 'improving cost competitiveness'. Revitalizing the African FTC market from producers' point of view requires 'informing coffee farmers about FTC'; 'government involvement'; 'solving the cartel and middleman problem'; and 'providing financial support'. Furthermore, to revitalize the African FTC from the consumers' point of view, this study found that strategies required include 'promoting and marketing the brand'; 'telling consumers FTC helps improve farmers' lives'; 'lowering the price'; 'improving accessibility'; and 'boosting the coffee consumption'.

The overall implication of this is that it is necessary to adopt market-based strategies to improve the FTC brand. First, to develop effective marketing and promotional measures for FTC, it is necessary to create an environment where not only coffee consumers but also coffee producers can learn more about, experience, and access FTC. As this study reveals, one of the reasons why Korean coffee consumers and African coffee consumers and producers do not consume or produce FTC is that they 'do not know much about it and have had little contact with it.' For Koreans, the lack of FTC awareness could also be because Koreans perceive little relevance or responsibility for poverty in Africa. Although Korea has entered the ranks of developed countries, it has grown economically only in a short period. As such, the awareness of ethical acceptance or social responsiveness is still low. Moreover, ethical consumption has not formed into a large market in Korea [41]. For Africans, the reason for the lack of awareness of FTC is that important information is limited to intellectuals and the political class due to high levels of poverty, the resulting high illiteracy rate and lack of education, electricity, and access to the Internet. These factors cut off the masses from information networks. Therefore, in order to revitalize the FTC market, it is most important to provide information to both Koreans and Africans through marketing, promotional activities and education. At the same time, it is vital to increase the accessibility of FTC in local stores. This is consistent with the interviewees' suggestions that they would purchase FTC if they were aware of it and if it was easily accessible.

Second, it is important to increase FTC's product competitiveness by matching the price to the quality. The most important coffee attributes for Korean and African coffee consumers are taste and reasonable price. Meanwhile, it is critical to ensure adequate profits from the coffee for the farmers. While ethical consumers may purchase FTC intermittently out of sympathy, or out of the fact that FTC aligns with their values, it is vital to build brand loyalty to stimulate repeat purchases. Andorfer and Liebe [56] performed a national field experiment on the actual purchase of FTC in three supermarkets in Germany, and among information, 20% price reduction, and a moral appeal, only price reduction had a positive effect on FTC consumption. This implies that price reduction could be the most effective way to promote the consumption of FTC. Similarly, Darian et al. [57] analyzed consumer motivation for purchasing FTC in America and found that improving wages and working conditions for workers and farmers were the main motivations. This suggests that for one to engage in ethical consumption, they need to see improvements in the farmers' economic conditions. The Coffee Association reported that, because FTC receives a fixed amount

of money, despite changes in international coffee beans prices, coffee farmers make more effort to produce specialty coffee beans, such as using better fertilizers and harvesting them manually, while less effort is put into producing FTC [57]. In this way, the FTC market is likely to continue to experience stagnation.

Finally, the benefits of FTC can only reach bona fide farmers when the exploitative middlemen prevalent in the coffee market, as well as political corruption related to coffee production, are eliminated. This will make FTC production appealing, thereby stabilizing FTC production and expanding the market size. Moreover, it will restore consumer confidence in the intentions of FTC and stimulate purchases among willing ethical consumers.

### 5.2. Theoretical Contribution and Practical Implications

The theoretical contribution of this study is that it has broken with the existing research trend in the field of business ethics, emphasizing that FTC is coffee with a good purpose in regard to ethical consumption. It is of academic significance in that it considered and compared FTC problems and ways to revitalize the FTC market, including cross-cultural factors relating to coffee consumers and producers among Koreans and Africans. These areas have not been examined in previous studies. In addition, by closely qualitatively analyzing the results of one-on-one interviews, it was possible to draw out the in-depth story of the research subjects, which is not well revealed by quantitative research.

The practical contribution of this study is on the revitalization of the FTC market. To achieve this goal, the study analyzed why producers and consumers have ignored FTC despite its good intentions. The paper then presented specific ways to improve the impact of FTC. To increase the recognition and accessibility of FTC, this study suggests that it is necessary to actively utilize global franchises, which have a global network, such as Starbucks and Nescafe. It is also necessary to improve the taste and quality of FTC to the level of specialty coffee in order to increase FTC competitiveness. Importantly, the study has revealed that the price of FTC is a major constraint to expanding FTC markets, not only in low-income countries in Africa, but also in a high-income country such as Korea. This problem has been largely ignored in previous studies. The study recommends that the FTC pricing problem should be addressed more concretely, not only in emerging coffee markets, but also in mature coffee markets such as Korea.

### 5.3. Limitations and Future Research

The limitation of this study is that the interviews were conducted among Africans currently living in Korea, and it is questionable how much they represented the position of African coffee producers as their social status may differ from rural African farmers. Moreover, if the African respondents' period of residence in Korea is prolonged, they can be immersed in the Korean coffee culture, which can lead to some differences in their understanding of African local coffee culture and FTC acceptance. However, due to the lack of electricity and internet in rural areas where African coffee farmers live, it was practically impossible to reach African coffee farmers for interviews. Additionally, face-to-face interviews with coffee farmers could not be possible because of the COVID-19 situation. To ensure that the interviews were well conversant with coffee farming issues in Africa, the researchers selected African interviewees who were born and raised in Africa, and who were educated in their own countries for more than 20 years. Some interviewees hail from coffee farming families or have relatives who are coffee farmers. Other interviewees had participated in projects related to coffee farming and stayed with coffee farmers in farm areas for several months. These measures limited the potential biases in interviewees' opinions and ensured the interviewees were well conversant with the lives of coffee farmers who live under poor circumstances, as well as coffee consumption habits in Africa.

Future studies should be conducted with local farmers in Africa who produce fair trade coffee. Where possible, face-to-face interviews with local coffee farmers could bring more insights into the study. It is also necessary to conduct a country-to-country study by comparing the FTC in coffee-producing countries. It is believed that more meaningful

results can be derived if the study is conducted in countries where the recognition and sales of FTC are actively carried out, such as the UK and the United States. Finally, an experiment on FTC awareness should be conducted to prove that attitudes toward FTC consumers could be changed if the awareness of FTC changes.

**Author Contributions:** Conceptualization, J.K. and M.J.; methodology, M.J.; software, M.J.; validation, M.J.; formal analysis, M.J.; investigation, M.J., J.K. and H.K.O.; resources, M.J.; data curation, M.J., J.K. and H.K.O.; writing—original draft preparation, M.J. and J.K.; writing—review and editing, J.K. and H.K.O.; visualization, M.J.; supervision, J.K.; project administration, J.K.; funding acquisition, J.K. All authors have read and agreed to the published version of the manuscript.

**Funding:** The paper was supported by the research grant of the University of Suwon in 2021 (grant number 2021-0006).

**Institutional Review Board Statement:** Not applicable.

**Informed Consent Statement:** Not applicable.

**Data Availability Statement:** Not applicable.

**Conflicts of Interest:** The authors declare no conflict of interest.

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
