# Peer review of "Why Are You Turning a Blind Eye to Fair Trade Coffee?—Focused on the Comparison between Korea and Africa"

_sustainability, doi:10.3390/su142417033_

Round 1
Reviewer 1 Report
Why are you turning a blind eye to fair-trade coffee? Focused on the comparison between Korea and Africa
Manuscript ID: sustainability-2030144
Thank you for the opportunity to review your manuscript. You did a good job and I have given some comments to further improve your manuscript. All the best for your research endeavors.
Introduction
The introduction needs change and focus. Some of the questions need to be answered in the introduction section.
i) Why only these two countries? Is it due to production (i.e., Africa) and consumption (i.e., Korea)?
ii) In the intro section there is no mention of what prior research has been carried out. The authors have mentioned that are only limited studies, but no specific study mentioned. The only citation is from the year 2013. What happens afterward?
iii) “This study aims to fill this gap by examining cross-cultural reasons why coffee consumers ignore fair trade coffee (FTC) which is considered ethical consumption. In addition, the study examines ways to revitalize the FTC market to promote ethical consumption in both Korea and Africa.”
Do the ordinary and regular consumers of coffee will think of FTC? The coffee consumption could be due to habit.
iv) Is there a parallel study that does in any other agricultural produce or commodities?
v) Why this research is important? For whom it is important? For policymakers or academia?
vi) To hook the readers, indicate your broad findings in the introduction.
vii) “Low awareness of fair-trade leads to a low purchase rate of fair-trade coffee.” (Page 2 line no 67) This assertion needs to be cited.
viii) Broadly, your introduction section needs the following:
a) What is the paper about?
b) What is known so far?
c) What issues are open so far?
d) What you have done?
e) What did you find?
2) Literature review: (i) the literature review has to align with the literature about the awareness level, and the questions of an in-depth interview. I request to rewrite.
(ii) Instead of Korean (2.1.1) or African market (2.1.2) – changes these titles and the content.
3) Research Methodology:
i) The Korean samples seem to be homogeneous when compared to the African samples.
ii) Why only 8 samples per country?
iii) What is the average time for each interview?
iv) How do you validate these interview questions?
Theoretical contribution – Need to rewrite and avoid the points which are meant for practitioners.
Cooperative advertisement – Possibly provide a few points about cooperative advertising about FTC. This will help to generate more awareness. For example (click the link) - https://brandequity.economictimes.indiatimes.com/news/advertising/neccs-egg-ceptional-campaign-and-the-story-bts/93221527.
Future study – Perform a casual experiment to prove the awareness of FTC.
Author Response
The title of our paper is “Why are you turning a blind eye to fair trade coffee? -Focused on the comparison between Korea and Africa” and was submitted to Sustainable Business Management and Cross-Culture Marketing Management Research on October 29, 2022. We have received valuable comments from three anonymous referees. Some of the feedbacks were common across the four referees, which were incredibly helpful in improving this work. We include our responses to major comments received by the four referees of Sustainability, to reflect how we have adapted their feedback to improve our work. We would appreciate the opportunity for our paper to be reconsidered for publication in Sustainable Business Management and Cross-Culture Marketing Management Research.

Reviewer 2 Report
Dear Authors,
There are many design problems with this research. I will describe below some of the issues:
1. The methods section should clearly state and justify why this method, for example, an interview, was chosen. The method was somehow outlined and illustrated with interview questions. Still, the criteria for selecting the study participants were not explained and justified. How the participants were recruited and by whom also must be stated. A brief explanation/description should be included of those who were invited to participate but chose not to. It is essential to consider “fair dealing,” i.e., whether the research design explicitly incorporates a wide range of different perspectives so that the viewpoint of 1 group (8 persons) is never presented as if it represents the only truth about any situation. The process by which ethical and or research/institutional governance approval was obtained should be described and accordingly cited.
2. Sampling is essential; unfortunately, I don’t see any description of the research setting. Sampling differs between qualitative and quantitative studies. Qualitative researchers should describe their sample in terms of characteristics and relevance to the broader population. Purposive sampling is typical in qualitative research. Individuals are chosen with features that are thought to be most informative to the study. The method for gaining informed consent from the participants should be described, as well as how anonymity and confidentiality of subjects were guaranteed. The method of recording, e.g., audio or video recording, should be noted, along with procedures used for transcribing the data.
3. The analytical approach should be better described and theoretically justified considering the missing research question. If more than one researcher repeated the analysis to ensure reliability or trustworthiness, this should be stated, and methods of resolving disagreements should be clearly described. Some researchers ask participants to check the data. If this was done, it should be thoroughly discussed in the paper.
4. Findings. Even though the researchers used NVivo R to explore and present the links between project items, there is no link with the literature in this field. Lack of cited works regarding the use of this software. An adequate account of how the findings were produced should be included. A description of how the themes and concepts were derived from the data also should be included. Was an inductive or deductive process used?
Author Response

(The authors gave the same response as above.)

Reviewer 3 Report
The Research is topic having Novelty and it's very well fit with Journal Scope as well .These are the area points that Authors can look in to
* Though the study is qualitative in nature , Proper Editing (of Response)is required which is missing in through out the paper especially in Discussion part
*Authors have interviewers Statements which needs to be Edited properly for a Scientific Research Article
* The basis of Selection of Samples are not Explained, Whether it is Judgemental sampling or Convenient or any other method
* I don't think the table one and table 2 are required in main text
* Discussion and findings parts , Editing is required , instead of respondent's using direct statement ,it has to be edited and presented in such a way that it should keep the standard of a good research article .
* Language Correction is required and can take the help of a native English speaker
Kindly look in to all these points
Author Response

(The authors gave the same response as above.)

Reviewer 4 Report
It is an excellent qualitative paper. The issue is relevant and timely. However, the argument of low awareness of FTC in lines 63-70 of about 7 per cent is premature. Could you provide further information on the growth of FTC over the last few years? The FTC growth trend could show the dynamic or awareness level of the FTC. It can also be argued that the percentage (7%) of awareness could be high if the base percentage of awareness is small, suppose 0.5 per cent for the last five years.
Author Response

(The authors gave the same response as above.)

Round 2
Reviewer 1 Report
Dear Authors,
The paper has developed a lot.
Reviewer 2 Report
I accept this revised version.
The authors improved the overall quality of the manuscript in terms of research methodology.
Reviewer 3 Report
Congratulations to Authors for carrying out the suggestion provided by all the reviewers and it's reflecting on the paper as well .